# GVote: Adaptive Per-request KV-Cache Compression without Manually Setting Budget

## Abstract

Large language models (LLMs) inference relies heavily on KV-caches to accelerate autoregressive decoding, but the resulting memory footprint grows rapidly with sequence length, posing significant efficiency challenges. Current KV-cache compression methods suffer from a Procrustes' bed problem: they force diverse workloads into fixed compression ratios, leading to suboptimal resource allocation and inference performance. To this end, we present GVote, an adaptive per-request KV-cache compression scheme that eliminates manual budget specification while achieving superior accuracy-efficiency trade-offs. GVote operates on the principle that the important keys are the aggregation of keys required by future queries. Gvote predicts future query attention demands by Monte-Carlo style sampling potential queries and aggregating selected keys to determine the optimal cache budget without manual specification. Experimental evaluation demonstrates GVote's effectiveness across multiple benchmarks, including GSM8K, RULER and Longbench. Compared to baselines, GVote exhibits $2\times$ memory reduction while the accuracy maintains higher or comparable.

## 1 Introducion

Large language models (LLMs) rely on key–value (KV) caches to store intermediate attention results during autoregressive decoding, enabling efficient reuse of previously computed representations. However, the deployment of LLMs faces a critical bottleneck: the quadratic growth of KV-cache memory with sequence length (Vaswani et al., 2017). In practical deployments like document summarization, RAG and question answering (Lewis et al., 2020; Raffel et al., 2020; Beltagy et al., 2020; Brown et al., 2020), the KV-cache often dominates GPU memory usage, limiting batch size, inflating inference latency, and restricting model accessibility in resource-constrained environments(Alizadeh et al., 2024; Liu et al., 2024). Consequently, efficient KV-cache compression has become essential for enabling long-context reasoning while maintaining the accuracy and efficiency of LLM inference. (Beltagy et al., 2020; Zaheer et al., 2020).

Recent advances such as SnapKV and AdaKV (Li et al., 2024; Feng et al., 2024) demonstrate that exploiting the sparsity of attention scores can yield substantial KV-cache compression. However, these approaches follow a rigid *fixed-budget* paradigm, where practitioners must pre-allocate a static memory quota (e.g., retaining 20% of the cache) without foreknowledge of the contextual demands of incoming requests. Such a *one-size-fits-all* design inevitably forces heterogeneous workloads into the same compression ratio, leading to a Procrustes' bed problem that creates an intractable trade-off between memory efficiency and model accuracy.

Consider a production LLM inference engine serving diverse requests, ranging from mathematical reasoning (e.g., GSM8K (Cobbe et al., 2021)) to long-document analysis (e.g., RULER-4K (Hsieh et al., 2024)) and various QA tasks. When the cache budget is set too low (e.g., 20%), memory efficiency is improved but reasoning-intensive tasks suffer catastrophic degradation, with accuracy dropping to nearly zero. Conversely, allocating a higher budget (e.g., 50%) preserves performance on complex tasks but results in substantial memory waste for simpler workloads, where accuracy remains stable even under lower budgets, as illustrated in Figure 1.

Fixed-budget KV-cache compression is fundamentally flawed for dynamic workloads. Any predefined budget risks either severe performance degradation or wasteful memory over-provisioning,

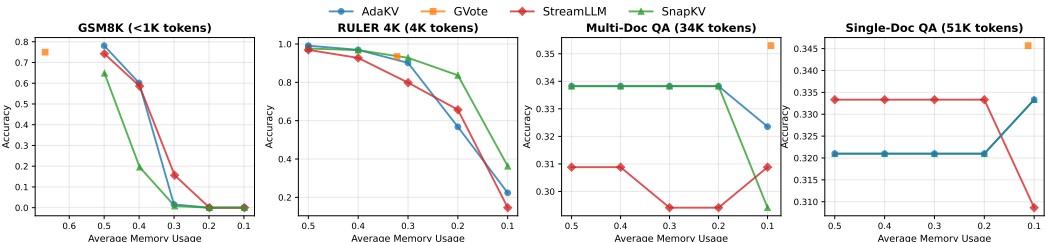

Figure 1: Accuracy-usage analysis across diverse benchmarks using the Llama-3.1-8B-Instruct model. The four panels display results for GSM8K (<1K tokens), RULER-4K (4K tokens), Multi-Doc QA (34K tokens) and Single-Doc QA (51K tokens), representing tasks with varying token lengths. The graphs show Accuracy on the y-axis against Average Memory Usage on the x-axis. The performance of StreamLLM, SnapKV, AdaKV, and GVote (ours) is compared. GVote consistently finds a favorable trade-off between accuracy and memory usage, achieving good accuracy while maintaining low memory consumption, thereby eliminating the need for manual budget settings, as shown in the plots.

a trade-off that necessitates costly, task-specific hyperparameter tuning (He et al., 2021) and creates brittleness to distribution shifts (Liu et al., 2021). This forces practitioners into conservative, inefficient memory allocation strategies.

We introduce GVote, a per-request adaptive compression scheme that eliminates manual budget tuning. Our approach stems from a key observation: LLM hidden states follow a discernible Gaussian distribution(NVIDIA, 2025). GVote leverages this by sampling a small number of representative states to dynamically compute the minimal required KV-cache for each request.

GVote consistently outperforms fixed-budget methods. Figure 1 starkly illustrates that different tasks have vastly different optimal compression ratios, rendering any single fixed budget inherently suboptimal. While static budgets force a compromise—sacrificing either memory on simple tasks or accuracy on complex ones—GVote automatically finds the sweet spot. For example, on Multi-Doc QA, GVote delivers 0.35 accuracy using only 10% average memory, surpassing baselines that use over 20% memory yet achieve lower accuracy. GVote thus breaks the rigid trade-off of fixed-budget methods, adaptively delivering both high performance and memory efficiency.

Our contributions are threefold. *First*, we identify and formalize the fundamental limitation of fixed-budget KV-cache compression in serving diverse workloads. *Second*, we propose GVote, a novel adaptive compression algorithm that automatically computes optimal cache budgets through query sampling and voting mechanisms. *Third*, we demonstrate substantial empirical improvements across multiple benchmarks, showing that adaptive computation significantly outperforms fixed-budget approaches while eliminating manual tuning requirements.

## 2 BACKGROUND

### 2.1 KV CACHE AND TOP-DOWN COMPRESSION

We consider modern autoregressive LLMs with $L$ layers and $H_{\mathrm{kv}}$ key-value heads, employing architectures like MHA (Vaswani et al., 2017) or GQA (Ainslie et al., 2023). Inference in these models is a two-stage process. First, during **prefilling**, the input prompt of length $S$ is processed to generate its Key-Value (KV) cache. Second, during **decoding**, the model generates tokens sequentially, where each new token's query attends to the KV cache of the prompt and all previously generated tokens.

The attention mechanism at each step $t$ computes weights $\mathbf{A}_t$ and aggregates values $\mathbf{V}_{1:t-1}$:

$$\mathbf{A}_t = \mathrm{softmax}\left(\frac{\mathbf{Q}_t\mathbf{K}_{1:t-1}^T}{\sqrt{d_k}}\right); \quad \mathrm{Attention}(\cdot) = \mathbf{A}_t\mathbf{V}_{1:t-1} \quad (1)$$

The memory footprint of the prompt's KV cache, generated during prefilling, scales linearly with $S$, creating a significant bottleneck for long contexts. Crucially, recent work has revealed that at-

tention weights $\mathbf{A}_t$ are often highly sparse, with only a few key-value pairs receiving most of the attention (Zhang et al., 2023; Li et al., 2024). This sparsity is the primary motivation for KV-cache compression: we can discard a large portion of the prompt's KV cache post-prefilling without substantially degrading performance.

**The Top-Down Compression Framework.** Existing methods that exploit this sparsity can be unified under a three-step, top-down framework applied to the prompt cache:

1. **Scoring:** An importance score $\mathbf{S} \in \mathbb{R}^{L \times H \times S}$ is computed for each token in the prompt's KV cache.

$$\mathbf{S} = \text{Score}(\mathbf{K}, \mathbf{V}, \mathbf{Q}). \tag{2}$$

2. **Allocation:** A pre-determined global budget $B$ is distributed into per-head quotas $b^{(\ell,h)}$.

$$\{b^{(\ell,h)}\} = \text{Alloc}(B, L, H, \mathbf{S}) \quad \text{s.t.} \quad \sum_{\ell,h} b^{(\ell,h)} = B. \tag{3}$$

3. **Selection:** Within each head, the top-$b^{(\ell,h)}$ tokens are retained based on their scores to form the compressed cache. During decoding, the prompt kv cache behaves as if it only has the kv cache of these tokens.

$$\mathcal{I}^{(\ell,h)} = \text{TopK}(\mathbf{S}^{(\ell,h)}, b^{(\ell,h)}). \tag{4}$$

All these methods follow a "cake-slicing" paradigm (Qin et al., 2025): a global budget $B$ is fixed *a priori*, and the subsequent steps merely select tokens to fit this size. The fundamental flaw lies in this static assumption. The optimal budget is highly dependent on the input's complexity and is unknowable at deployment time. This forces a dilemma: either over-provision memory for simple prompts, wasting resources, or risk catastrophic performance degradation on complex ones by setting the budget too low. Our work directly addresses this limitation.

## 2.2 EXISTING WORKS

**Scoring Strategies.** A first line of work focuses on designing token-importance *scores* from which the cache is pruned. For example, StreamingLLM (Xiao et al., 2023) leverages a sliding-window augmented with attention sinks to approximate query–key affinity. SnapKV (Li et al., 2024) selects important key-value pairs by analyzing the attention from a small tail window of tokens to the prompt, retaining those with high cumulative attention. H2O (Zhang et al., 2023) formulates the selection of important key-value pairs as a heavy hitter detection problem, leveraging algorithms from large-scale data analysis. In practice, these works commonly apply a Top-K rule to select tokens based on their computed scores.

**Allocation Policies.** A complementary line of research studies how a global memory budget should be *allocated*. The simplest heuristic is a *global Top-k* rule that keeps the highest-scoring tokens irrespective of their origin. More refined schemes assign budgets at the *layer* level—PyramidKV (Cai et al., 2024) uses heuristics to distribute the budget across layers, while SimLayerKV (Zhang et al., 2024) leverages defined layer similarity to guide allocation. At the *head* level, AdaKV (Feng et al., 2024) achieves theoretical optimality in allocation at the attention score level given a fixed budget, though this does not always translate to optimal end-to-end performance. These works make important contributions to the allocation problem, but they also exacerbate the fundamental challenge: *how should the budget be determined in the first place?*

## 3 METHOD

GVote departs from the top-down "cake-slicing" paradigm. Instead of compressing within a fixed budget, it adaptively *determines the budget itself* based on a bottom-up estimation of future requirements. This section first establishes the core hypothesis of our approach—that synthetic queries can serve as a useful, albeit imperfect, proxy for future ones—and provides empirical validation. Then, grounded in this nuanced understanding, we detail the GVote algorithm, including a key design choice motivated by the proxy's limitations.

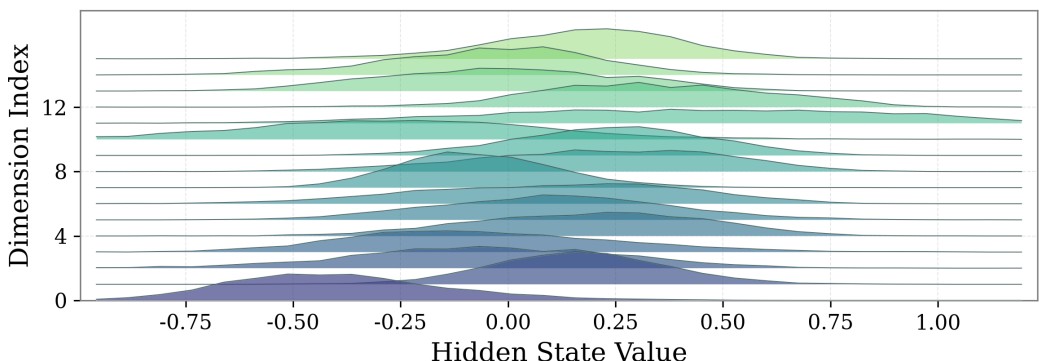

Figure 2: Hidden states illustration using Qwen3-1.7B at head 12 of layer 12. They exhibit explicit channel-wise Gaussian structure.

## 3.1 MOTIVATION

What defines an optimal KV-cache compression? We argue it is one that preserves the minimal set of keys whose attention weights constitute the **top-$p$ cumulative probability mass**. This criterion provides a natural trade-off, retaining only the most influential tokens for each query.

Imagine an oracle that knows all future queries $\{\mathbf{Q}_t, \mathbf{Q}_{t+1}, \ldots\}$ that will be generated. The optimal compression strategy would be to keep the *union* of the top-$p$ key-sets required by each of these future queries. Formally, the ideal keep-set is $\mathcal{K}^* = \bigcup_{u \geq t} \arg \text{Top-p}(\mathbf{A}_u)$, where $\mathbf{A}_u$ is the attention map for a future query $\mathbf{Q}_u$. This oracle perspective reveals the fundamental goal: the compressed cache must satisfy the *aggregate needs* of all subsequent decoding steps.

The practical challenge, of course, is that future queries are unknown. How can we estimate this ideal set $\mathcal{K}^*$? Our key insight stems from a statistical regularity in Transformers: hidden states, when viewed across the sequence dimension, approximate a Gaussian distribution (NVIDIA, 2025). This property, which we verify in Figure 2, arises from operations like Layer Normalization (Ba et al., 2016; Sun et al., 2024).

This finding provides a principled way to approximate the unknown future. By sampling from a Gaussian distribution fitted to the current hidden states, we can generate a set of *plausible future queries*. The union of the key-sets required by these synthetic queries can then serve as a robust proxy for the ideal keep-set $\mathcal{K}^*$. This Monte Carlo approach, where multiple synthetic queries "vote" on which keys to retain, is the foundation of GVote.

## 3.2 VALIDATION: ARE SYNTHESIZED QUERIES GOOD PROXIES?

Before building an algorithm upon this idea, we must critically evaluate its central premise. We compare the attention patterns from our synthetic queries against those from real, ground-truth subsequent queries. To quantify this comparison, we define *Attention Overlap* as follows: for each synthesized query, we select the tokens corresponding to the top-0.95 cumulative attention weights to form a candidate set. The overlap score is then computed as the sum of attention weights in the ground-truth query that correspond to the same tokens in this candidate set. This metric quantifies how well our synthesized queries capture the salient tokens that the model actually attends to in the future. We also measure the corresponding token usage rate; and visualize a specific head's attention pattern and illustrate the correspondence of real attention weights and synthetic attention weights in Figure 3.

We observe a strong positive correlation ($r = 0.7759$) and a high attention overlap ($\mu = 0.929$), which validates that synthetic queries are indeed a meaningful proxy for real future queries. They successfully capture the general trend of where future attention will be directed.

However, the analysis also reveals crucial imperfections. The correlation, while strong, is not perfect, and the scatter plot shows the presence of outliers. This indicates that a single synthetic query

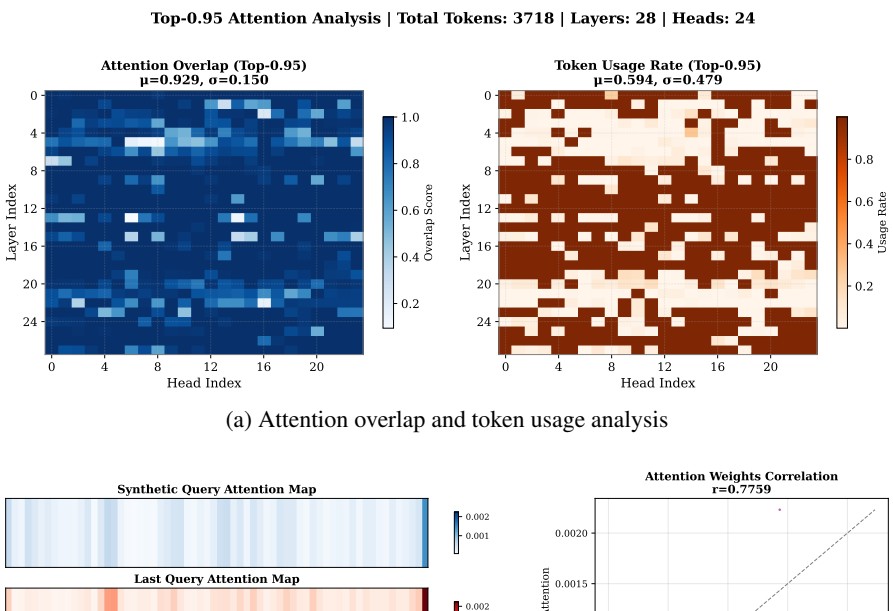

(a) Attention overlap and token usage analysis

(b) Attention correlation analysis

Figure 3: Comprehensive analysis of synthetic query approximation quality. (a) Layer-wise and head-wise attention overlap and token usage rates, showing consistency between synthetic and real query attention patterns with high overlap scores (mean $\mu = 0.929$) and moderate token usage rates (mean $\mu = 0.594$). (b) Detailed attention correlation analysis for a specific attention head, including attention map comparison and correlation scatter plot with $r = 0.7759$, demonstrating strong alignment between synthetic and real query attention patterns.

can occasionally produce a noisy attention distribution where some token scores are significantly over- or under-estimated compared to the ground truth. Specifically, oracle-style "top-p" is sensitive to outliers and it urges us to seek for another robust selection methods . For instance, under a top-0.95 criterion, if a single high-probability token's score is underestimated (e.g., from a true $0.8$ to a predicted $0.6$), the residual probability mass required to meet the threshold more than doubles (from $0.15$ to $0.35$). This gap must then be filled by a long tail of low-probability, often noisy, tokens, causing the selection set to expand dramatically. This motivates a more robust selection mechanism.

This motivates a more robust selection mechanism. We observe that top-$p$ selection is effectively a method to dynamically determine a $K$ for a top-$k$ operation, and its instability results in a poor estimation of real $K$. To mitigate this, we can adopt a more stable top-$k$ approach. We propose using the last ground-truth query as a stable anchor: we compute the $K$ it would require to meet the top-$p$ threshold and apply this fixed $K$ as an estimated selection size for all subsequent synthetic queries in the step. This estimated $K$ is often significantly smaller than the true number of tokens that would ultimately need to be retained for optimal performance. This is precisely what we aim for, as we can refine our estimation by increasing the number of synthetic samples ($S$), avoiding the pitfall of over-selecting noisy tokens from the outset.

## 3.3 THE PROPOSED GVOTE SCHEME

To counteract the noise sensitivity of a single synthetic query, we propose **GVote (Gaussian-based Voting)**, a robust KV-cache compression scheme based on Monte Carlo aggregation. As detailed in

---

**Algorithm 1** GVote: Adaptive Monte–Carlo KV-cache compression

---

**Require:** Keys $\mathbf{K}_{1:L} \in \mathbb{R}^{H_k \times L \times d_k}$, values $\mathbf{V}_{1:L}$, hidden state $\mathbf{h} \in \mathbb{R}^{L \times d_h}$, nucleus threshold $p_{\text{nuc}}$, number of samples $S$, number of future positions $n_f$

1: $\mathbf{A}_0 \leftarrow \text{softmax}\big(\mathbf{Q}_0 \mathbf{K}_{1:L}^\top / \sqrt{d_k}\big)$       // $\mathbf{Q}_0$ is current query, $\mathbf{A}_0 : \mathbb{R}^{1 \times L}$
2: $\mathcal{C}_0 \leftarrow \text{TopP}\big(\mathbf{A}_0, p_{\text{nuc}}\big)$
3: $B_{\text{step}} \leftarrow |\mathcal{C}_0|$                // step budget
4: $\boldsymbol{\mu} \leftarrow \text{mean}(\mathbf{h})$, $\boldsymbol{\sigma}^2 \leftarrow \text{var}(\mathbf{h})$          // Step 2
5: $\widetilde{\mathbf{H}} \sim \mathcal{N}(\boldsymbol{\mu}, \boldsymbol{\sigma})$             // $\widetilde{\mathbf{H}} \in \mathbb{R}^{S \times d_h}$
6: $\mathbf{P} \leftarrow \text{AverageCosSin}(n_f)$     // Compute average cos/sin for $n_f$ future positions
7: $\widetilde{\mathbf{Q}} \leftarrow \text{Reshape}\big(\text{RoPE}(\widetilde{\mathbf{H}}\mathbf{W}_q, \mathbf{P})\big)$    // Apply RoPE using averaged positions, $\widetilde{\mathbf{Q}} \in \mathbb{R}^{H_q \times S \times d_k}$
8: $\mathbf{L} \leftarrow \widetilde{\mathbf{Q}} \mathbf{K}_{1:L}^\top / \sqrt{d_k}$        // attention logits, $\mathbf{L} \in \mathbb{R}^{H_q \times S \times L}$
9: $\{\mathcal{C}^{(s)}\}_{s=1}^S \leftarrow \text{TopK}\big(\mathbf{L}, B_{\text{step}}\big)$       // row-wise on logits
10: $\mathcal{K} \leftarrow \bigcup_{s=1}^S \mathcal{C}^{(s)}$            // Step 4
11: Prune $(\mathbf{K}_{1:L}, \mathbf{V}_{1:L})$ to the indices in $\mathcal{K}$
12: **return** Compressed KV-cache $(\mathbf{K}_\mathcal{K}, \mathbf{V}_\mathcal{K})$

---

Algorithm 1, the procedure first establishes a dynamic selection budget, $B_{\text{step}}$, by applying nucleus sampling to the current query's attention scores. It then synthesizes $S$ diverse future queries by sampling from a channel-wise Gaussian distribution fitted to the model's hidden states. Each of these queries "votes" for its top-$B_{\text{step}}$ tokens from the cache. The final keep-set, $\mathcal{K}$, is formed by the union of all votes, effectively preserving tokens deemed important across multiple plausible futures while filtering out noise from any single prediction. The entire process is performed per-request and per-head, adapting naturally to heterogeneous workloads.

### 3.4 Implementation and Overhead

The GVote algorithm is designed for efficient, parallel execution on GPU architectures. The $S$ synthetic queries are processed in a single batch, amortising the computational cost. The union operation (line 01) is performed efficiently using boolean masks (`torch.any`). And the final indices for $\mathcal{K}$ are extracted via a `nonzero` operation. The subsequent pruning (line 11) is then a simple indexing operation which gathers the selected keys and values. This entire procedure introduces a one-time computational overhead during the prefill phase. While non-trivial, this cost is fixed and does not scale with the number of generated tokens.

During the decoding phase, the GVote procedure is applied independently for each attention head, leading to a *non-uniform* KV-cache where different heads may retain a varying number of key-value pairs. This structurally sparse format precludes storage as a single dense tensor. However, it is fully compatible with modern attention kernels like FlashAttention (Dao, 2023), which are designed to handle variable-length sequences via the `varlen` interface. This approach of creating a non-uniform, content-aware cache is consistent with prior work like AdaKV (Feng et al., 2024). We provide a detailed PyTorch-style implementation of GVote in Appendix A.

### 3.5 Parameters

The GVote algorithm introduces two key hyperparameters: the nucleus sampling threshold $p_{\text{nuc}}$ and the number of synthetic queries $S$.

**Nucleus threshold ($p_{\text{nuc}}$).** This parameter (Algorithm 1, line 2) sets the cumulative probability mass of attention weights that must be preserved—effectively specifying the recall of the softmax attention distribution. We recommend $p_{\text{nuc}} = 0.95$ in practice, which offers decent accuracy-memory trade-off. For short sequences $p_{\text{nuc}} = 0.95$ would suggests a slightly larger proportion, yet the effect is trivial since the absolute length of the sequence is small. Detailed sensitivity analysis is presented in Section 4.5.

**Number of samples ($S$).** This parameter (Algorithm 1, line 5) controls the number of synthetic future queries used to form the final keep-set $\mathcal{K}$. In practice, the impact of $S$ on the final budget is less pronounced than that of $p_{\text{nuc}}$. While a larger $S$ can produce a more robust estimate, it typically

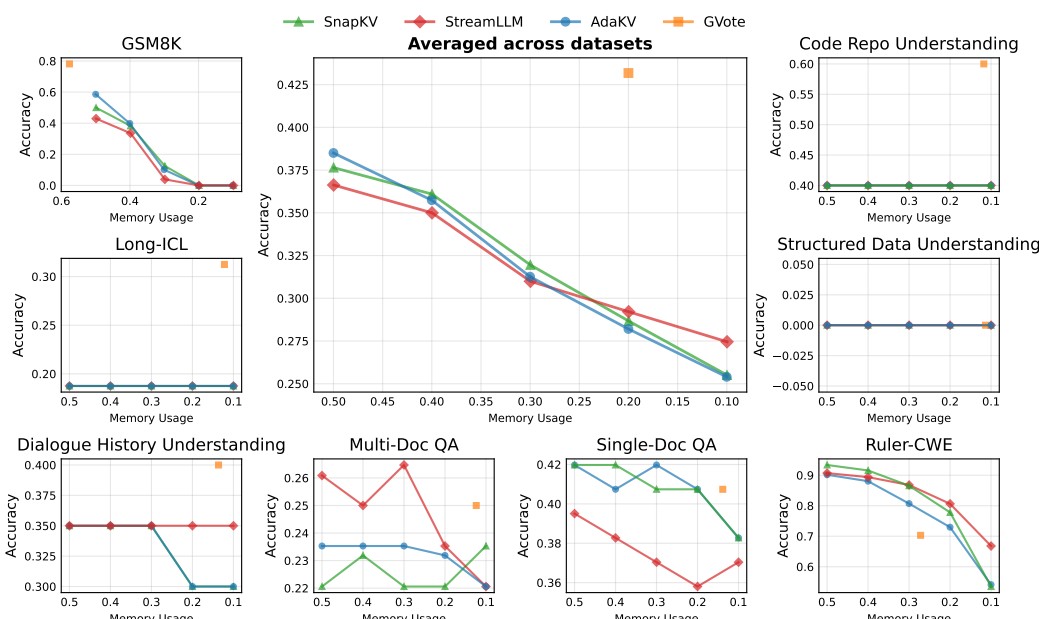

Figure 4: Accuracy vs. Cache Usage across eight benchmarks using Qwen2.5-7B-Instruct. Each baseline shows results across different compression ratios (10%-50%). The optimal budgets across different datasets are various, a characteristics that fixed budget cannot deal with. GVote (orange square) consistently achieves accuracy-usage trade-off sweet spot compared to baselines with fixed budgets.

comes with higher KVCache usage and higher computational overhead. We find that $S \geq 8$ typically offers a good compromise between robustness and efficiency. We recommend using a conservative $p_{\text{nuc}}$ with a moderately high $S$. Further empirical results are discussed in Section 4.5.

## 4 EXPERIMENTS

### 4.1 EXPERIMENTAL SETUP

**Datasets.** We evaluate on eight diverse benchmarks spanning different sequence lengths and task types: GSM8K (mathematical reasoning)Cobbe et al. (2021), RULER-CWE Hsieh et al. (2024), and LongbenchBai et al. (2024) (long-context understanding).

**Baselines.** We compare against three representative KV-cache compression methods: **Stream-LLM** Xiao et al. (2023), **SnapKV** Li et al. (2024), and **AdaKV** Feng et al. (2024).

**Model and Implementation.** We extensively tests four popular models accross different families and sizes: Llama3.1-8B-Instruct, Llama3.2-3B-InstructDubey et al. (2024), Qwen2.5-7B-Instruct, Qwen2.5-14B-Instruct Team (2024). All experiments are conducted on a single NVIDIA RTX-A6000 (NVIDIA, 2025).

**Evaluation Metrics.** We measure accuracy and the effective cache usage ratio. For each baseline, we sweep compression ratios from 10% to 50% to construct accuracy-usage curves. GVote operates without a budget specification, automatically determining the optimal compression ratio per request.

### 4.2 MAIN RESULTS

Figure 4 presents the core results across all eight datasets and reports the average performance. GVote achieves higher accuracy under significantly low budget under most scenarios, and hence the overall performance is impressive, reducing 2x memory budget while maintaining the accuracy.

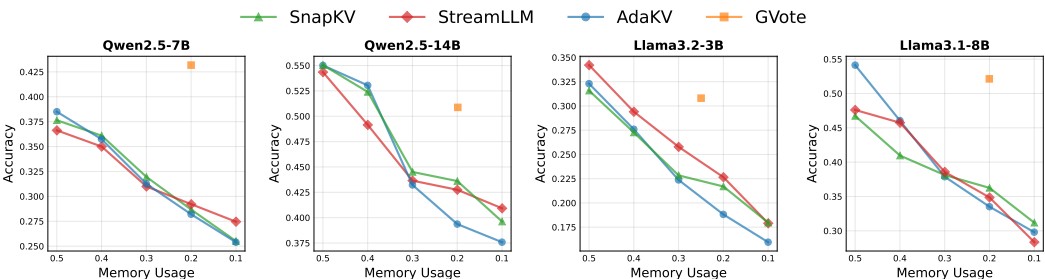

Figure 5: GVote performance across different model architectures and sizes. All models are of Instruct version. We omit the suffix due to space limitation. The figure shows accuracy-usage curves for various models, demonstrating the adaptive mechanism's generalization across different model configurations.

### 4.3 Multi-Model Evaluation

Figure 5 demonstrates GVote's effectiveness across different model architectures and sizes. The adaptive compression mechanism maintains its advantages across various model configurations, indicating good generalization properties. Full evaluation matrices are presented in Appendix B.

### 4.4 Latency and Overhead Analysis

We analyze GVote's computational overhead in the prefill and generation phases (Figure 6) using a synthetic dataset for precise length with Llama-3.1-8B-Instruct. During prefilling, the overhead of compression is noticeable for short contexts, where GVote's latency is modestly higher than other compression methods (Figure 6a). However, as context length increases, model computation becomes the dominant bottleneck, rendering the relative overhead of GVote's logic negligible. This ensures GVote's scalability for long-context scenarios where its benefits are most critical.

In the generation phase, we evaluate latency under a fair comparison framework where each method is configured to match the accuracy of GVote using Llama-3.1-8B-Instruct. The reported latency in Figure 6b represents the computational time required for each method given the memory usage needed to achieve this target accuracy, with details presented in Appendix C. GVote consistently demonstrates the lowest latency across all context lengths, from 2k to 128k. This superior efficiency is primarily due to its lower memory requirements; by maintaining high accuracy with a smaller memory footprint, GVote inherently reduces the computational overhead during token generation. While the latency of all methods increases with context length, the overhead introduced by other methods like AdaKV, SnapKV, and particularly StreamLLM, grows substantially more pronounced. In contrast, GVote's latency remains significantly lower, showcasing its superior performance and scalability for low-latency generation, especially in long-context scenarios where efficient memory management is critical.

### 4.5 Hyperparameter Analysis

In this section, we investigate about what is the optimal hyperparameters of GVote. We primarily test different parameter combinations on RULER dataset.

**Sampling number $S$.** Figure 7 demonstrates the effects of different sampling number $S$ on final performance. As intuition, higher $S$ would approximate the real distribution better. However, in case the context length is huge, for example, $> 128K$, higher $S$ would result in an extremely large intermediate logits matrix $\mathbf{L}$, potentially leading to out of memory errors.

**Nucleus Sampling Threshold ($p_{nuc}$).** Figure 8 examines how the nucleus sampling threshold for single-step budget estimation affects overall performance. Similar to $S$, higher $p_{nuc}$ would also result in both high accuracy and memory usage. However, it is more sensitive to $S$, as it reaching comparable accuracy with higher memory. The specific value of $p_{nuc}$ has no impact on pruning speed, unlike $S$.

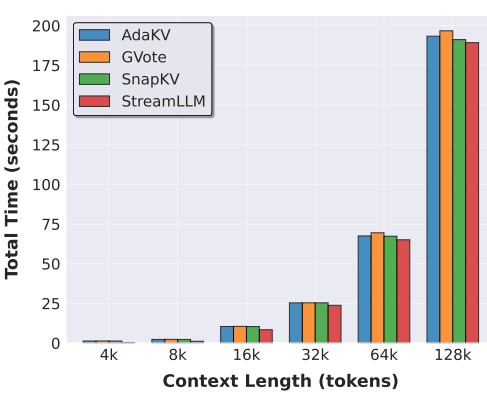

(a) Total prefill time vs. context length.

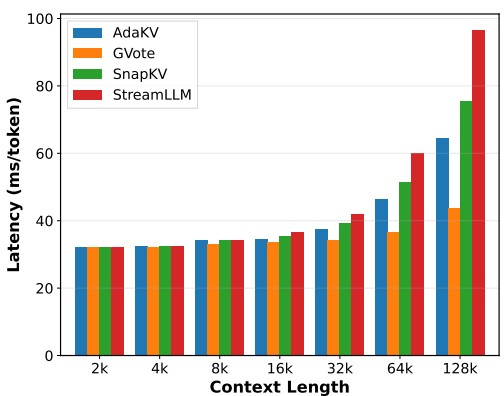

(b) Per-token generation latency (ms/token) bar-chart.

Figure 6: Latency analysis for the prefill and generation phases. (a) shows GVote's prefill overhead is comparable to other methods or insignificant compared to the overall prefill time. (b) demonstrates GVote's superior generation latency, especially when context length is large.

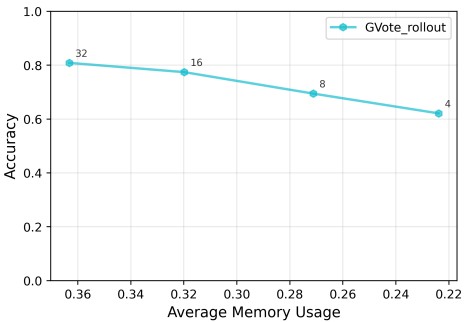

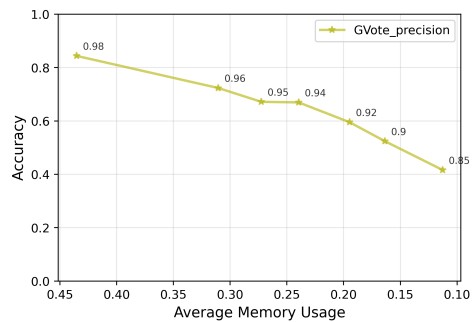

Figure 7: Effect of sampling number $S$ on GVote performance. Higher $S$ would deliver better accuracy with moderately higher usage. However it would cost more memory and computation for compression.

Figure 8: Effect of nucleus sampling threshold $p_{nuc}$. Similarly, higher $p_{nuc}$ would result in a higher accuracy and memory usage. However, it seems to be more sensitive compared to $S$.

## 5 CONCLUSION

We observe the uneven budget requirements of different requests and hence propose GVote to address this challenge. By moving beyond a *one-size-fits-all* approach and embracing the heterogeneous sparsity of requests through a Gaussian distribution, we have shown a clear path to substantial memory reduction. Our findings open the door for exciting future work. Specifically, we aim to investigate even more sophisticated adaptive methods that eliminate the need for an accuracy-memory trade-off. We also look forward to tackling the critical engineering challenges required to bring these theoretical gains into practical application.

## 6 LLM USAGE

The ideation, testing, and research process for this paper were human-led, with assistance from a Large Language Model (LLM) for refinement. Specifically, the manuscript was initially drafted by humans, after which an LLM was used for grammar and formatting checks. Furthermore, the experimental data was collected manually by humans, while its visualization was created with the assistance of the LLM.

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

# A   IMPLEMENTATION DETAILS

In this section, we provide a detailed PyTorch-style implementation of the GVote algorithm, as outlined in Algorithm 1. The following code (Listing 1) demonstrates the core logic for generating a KV-cache pruning mask. The implementation is fully vectorized to ensure high efficiency on GPUs and is designed to be compatible with modern transformer architectures, including a precise handling of Grouped-Query Attention (GQA).

The function gvote_get_mask takes as input the necessary tensors and configuration parameters and returns a boolean mask. This mask identifies the key-value pairs that are deemed most important for future token generation and should be retained in the cache. The implementation faithfully follows the four main steps of the GVote algorithm:

1. **Adaptive Budget Calculation:** It first determines an independent budget of KV entries to keep ($B_{step}$) for each of the model's query heads by applying nucleus sampling (Top-P) to the attention scores of the current query.

2. **Hypothetical Query Generation:** It then models future hidden states by sampling from a normal distribution parameterized by the mean and variance of the existing hidden states. These are projected into hypothetical future queries and infused with averaged future positional information via RoPE.

3. **Importance Voting:** Each of the hypothetical queries attends to the entire KV-cache. We then select the most important tokens for each query head based on its unique budget calculated in the first step. To achieve this efficiently, we use a rank-based selection method (double argsort) instead of a conventional topk operation, which allows for a different number of tokens to be selected for each head in a fully vectorized manner.

4. **Mask Creation and Union:** A boolean selection mask is created for each query head. For models using GQA, the masks from query heads within the same group are combined via a logical OR (torch.any) operation, effectively taking the union of their votes. Finally, a second union operation is performed across all hypothetical samples to produce the pruning mask for the entire KV-cache.

```python
import torch
import torch.nn.functional as F
import math

def gvote_get_mask(
    q_current: torch.Tensor,
    k_cache: torch.Tensor,
    hidden_states: torch.Tensor,
    rotary_emb: torch.nn.Module,
    q_proj_weight: torch.Tensor,
    p_nuc: float,
    num_samples: int,
    num_future_positions: int,
    num_kv_heads: int,
) -> torch.Tensor:
    """
    PyTorch-style implementation of GVote (Algorithm 1) to generate a
    pruning mask.
    """
    # Get input shapes
    batch_size, num_q_heads, _, head_dim = q_current.shape
    _, _, seq_len, _ = k_cache.shape
    hidden_dim = hidden_states.shape[-1]
    device = k_cache.device
    num_groups = num_q_heads // num_kv_heads

    # === Step 1: Adaptive Budget (Lines 1-3) ===
    k_cache_rep = k_cache.repeat_interleave(num_groups, dim=1)
    attn_scores = torch.matmul(q_current, k_cache_rep.transpose(-2, -1))
    / math.sqrt(head_dim)
```

```
29    attn_probs = F.softmax(attn_scores, dim=-1).squeeze(2)
30
31    sorted_probs, _ = torch.sort(attn_probs, dim=-1, descending=True)
32    cumulative_probs = torch.cumsum(sorted_probs, dim=-1)
33    # Calculate a different budget b_step for each query head. Shape: [B,
       H_q]
34    b_step_per_head = torch.sum(cumulative_probs < p_nuc, dim=-1) + 1
35
36    # === Step 2: Generate Hypothetical Future Queries (Lines 4-7) ===
37    mu = torch.mean(hidden_states, dim=1)
38    std = torch.std(hidden_states, dim=1)
39
40    h_tilde = mu.unsqueeze(1) + std.unsqueeze(1) * torch.randn(
41        batch_size, num_samples, hidden_dim, device=device
42    )
43
44    q_tilde = F.linear(h_tilde, q_proj_weight)
45    q_tilde = q_tilde.view(batch_size, num_samples, num_q_heads, head_dim
       ).transpose(1, 2)
46
47    future_positions = torch.arange(seq_len, seq_len +
       num_future_positions, device=device)
48    cos, sin = rotary_emb.forward(future_positions)
49    avg_cos = cos[None, :, :].mean(dim=1, keepdim=True)
50    avg_sin = sin[None, :, :].mean(dim=1, keepdim=True)
51
52    def apply_rotary_pos_emb(q, cos, sin):
53        q_embed = (q * cos) + (torch.cat([-q[..., 1::2], q[..., ::2]],
       dim=-1) * sin)
54        return q_embed
55
56    q_tilde_rope = apply_rotary_pos_emb(q_tilde, avg_cos, avg_sin)
57
58    # === Step 3 & 4: Vote for Important Keys and Create Mask (Lines
       8-15) ===
59    logits = torch.matmul(q_tilde_rope, k_cache_rep.transpose(-2, -1)) /
       math.sqrt(head_dim)
60
61    ranks = logits.argsort(dim=-1, descending=True).argsort(dim=-1)
62
63    # Create selection mask for each query head based on its unique
       budget
64    # Shape: [B, H_q, S_samples, S_len]
65    selection_mask = ranks < b_step_per_head.unsqueeze(-1).unsqueeze(-1)
66
67    # If using GQA, combine masks within each KV group using a logical OR
        (union)
68    if num_groups > 1:
69        selection_mask = selection_mask.view(
70            batch_size, num_kv_heads, num_groups, num_samples, seq_len
71        )
72        # Union within the group: keep if ANY query head in the group
       voted for it
73        mask_per_kv_head = torch.any(selection_mask, dim=2)
74    else:
75        mask_per_kv_head = selection_mask
76
77    # Union across the hypothetical samples: keep if ANY sample voted for
        it
78    # Shape: [B, H_k, S_len]
79    final_mask = torch.any(mask_per_kv_head, dim=2)
80    return final_mask
```

Listing 1: Efficient, vectorized PyTorch-style implementation of GVote with precise GQA handling.

## A.1 INFERENCE WITH EVICTED CACHE

Once the pruning mask has been generated by the GVote algorithm, it must be efficiently applied during the attention computation of subsequent generation steps. A naive approach of physically rearranging the KV-cache for each head would incur significant memory movement overhead. To circumvent this, we leverage specialized, highly optimized attention kernels designed for variable-length sequences, specifically the `flash_attn_varlen_func` provided by the FlashAttention library (Dao, 2023).

The core strategy is to treat the pruned KV-cache of each head as an independent sequence of variable length. The entire process can be conceptualized in the following steps:

1. **Physical Cache Compaction:** Given the boolean eviction mask $\mathcal{M}$ of shape $[B, H_k, S_{\text{len}}]$, we first compact the key and value caches. This is achieved by flattening both the cache and the mask tensors and then using the mask for boolean indexing. This operation produces new, dense tensors containing only the key-value pairs marked for retention, concatenated together.

2. **Calculating Per-Head Lengths:** After compaction, the KV-cache for each head has a new, shorter length. This length is easily calculated by summing the boolean mask along its sequence length dimension ($L_i = \sum_j \mathcal{M}_{i,j}$). This results in a tensor of shape $[B \times H_k]$ containing the precise length of the retained KV-cache for every head.

3. **Constructing Cumulative Sequence Lengths:** The most critical step is to construct the cumulative sequence length tensor, `cu_seqlens_k`. This tensor is the cumulative sum of the per-head lengths, with an initial zero prepended. For instance, if the lengths of three heads are $[l_0, l_1, l_2]$, the `cu_seqlens_k` tensor will be $[0, l_0, l_0 + l_1, l_0 + l_1 + l_2]$. This tensor acts as an index, informing the kernel where the data for one head ends and the next begins within the compacted tensor.

4. **Executing Variable-Length Attention:** Finally, the compacted key and value caches, along with the query states and the newly created `cu_seqlens_k` tensor, are passed to the `flash_attn_varlen_func`. The kernel interprets this input as a batch of $B \times H_k$ sequences with varying lengths and computes the attention output with maximum efficiency.

This methodology allows the attention computation to proceed directly on the compacted data structure, seamlessly integrating the cache eviction process with the highly optimized forward pass of the attention layer.

## B DETAILED MULTI-MODEL EVALUATION RESULTS

In this section, we report the detailed evaluation matrices of Qwen (Figure 9) and Llama (Figure 10) families. The results of GVote are reported based on $p_{nuc} = 0.95, 0.9, 0.85$ and $S = 8$.

## C GENERATION HEATMAP

Figure 11 shows the generation latency of Llama-3.1-8B-Instruct on A6000, across various context lengths.

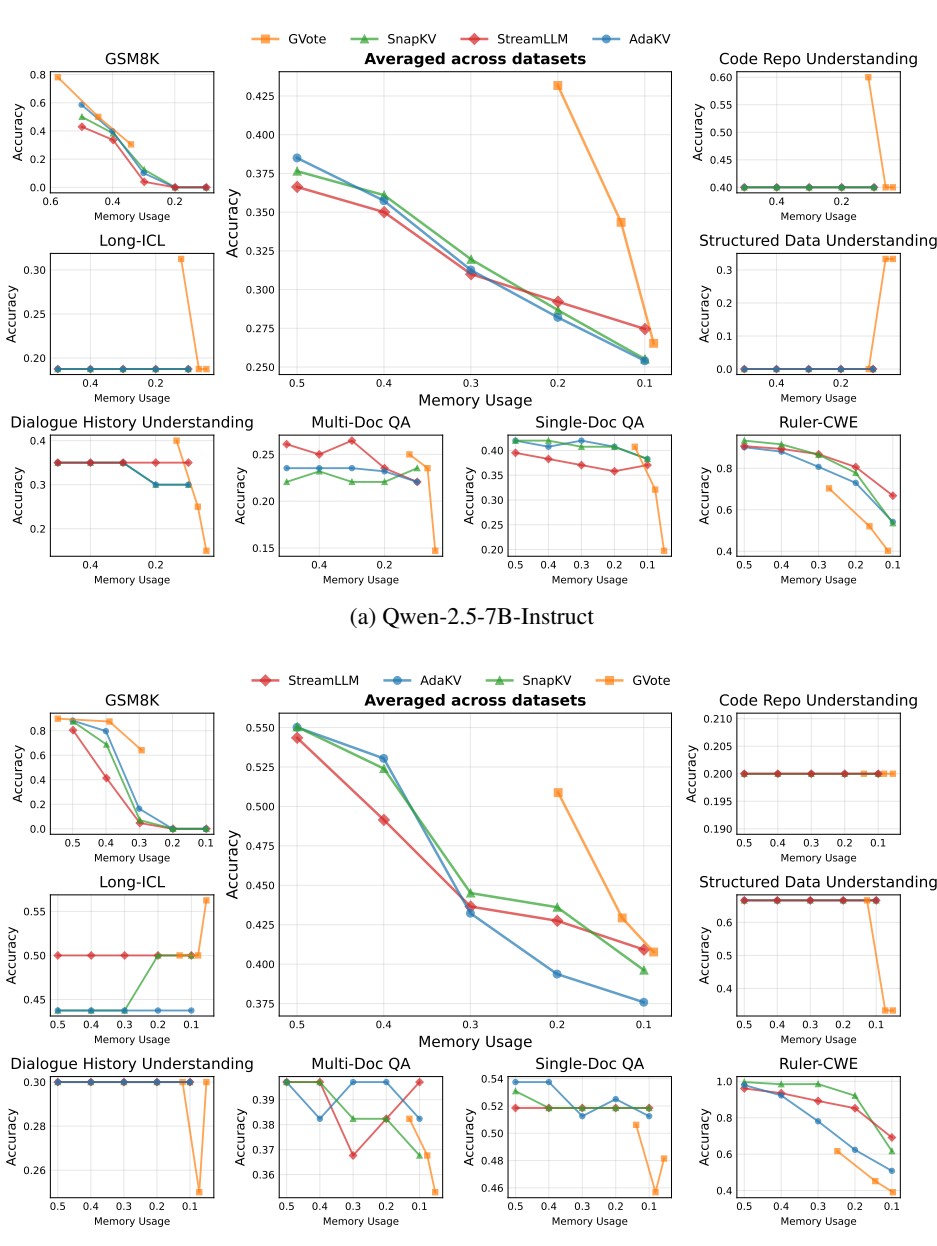

(a) Qwen-2.5-7B-Instruct

(b) Qwen-2.5-14B-Instruct

Figure 9: Metrics for the Qwen models.

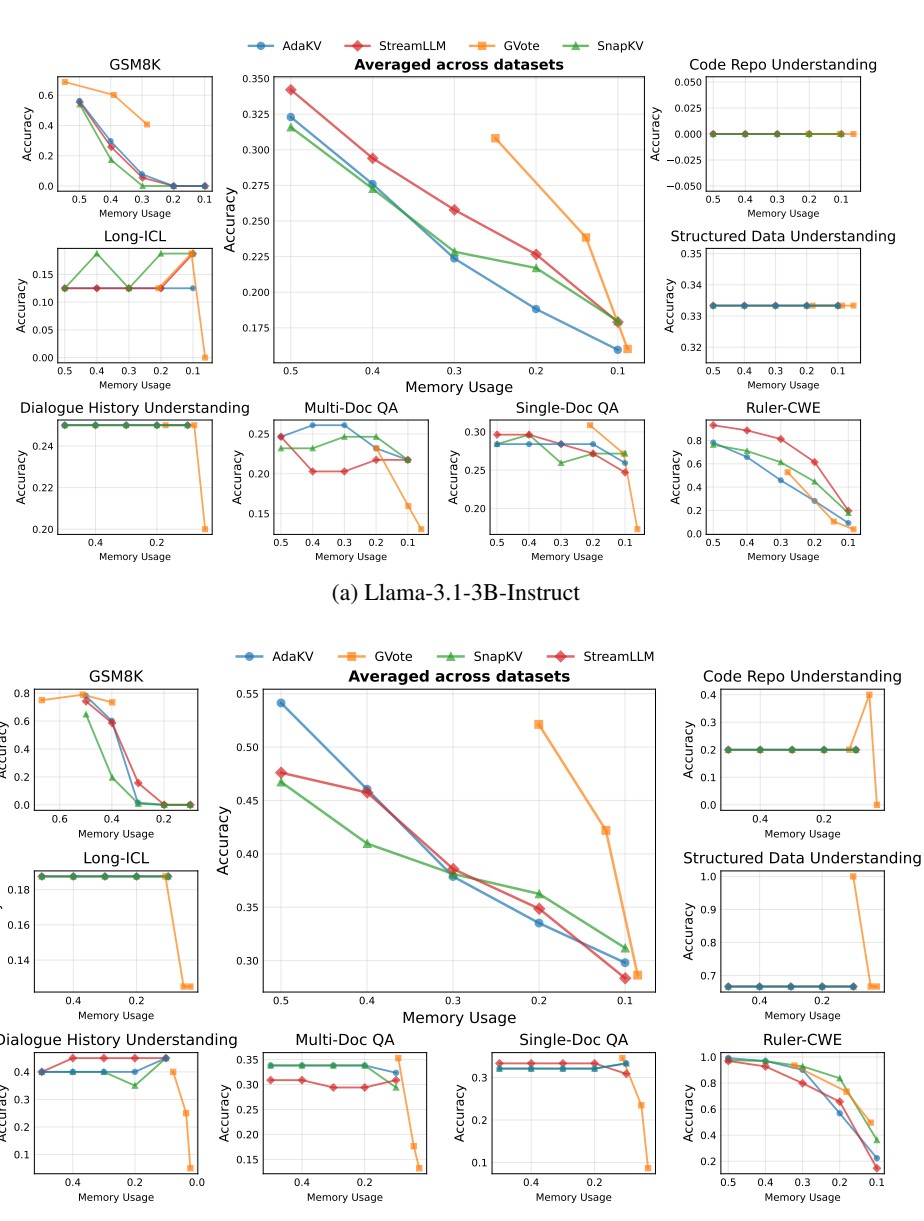

(a) Llama-3.1-3B-Instruct

(b) Llama-3.1-8B-Instruct

Figure 10: Accuracy-Usage plots for the Llama models.

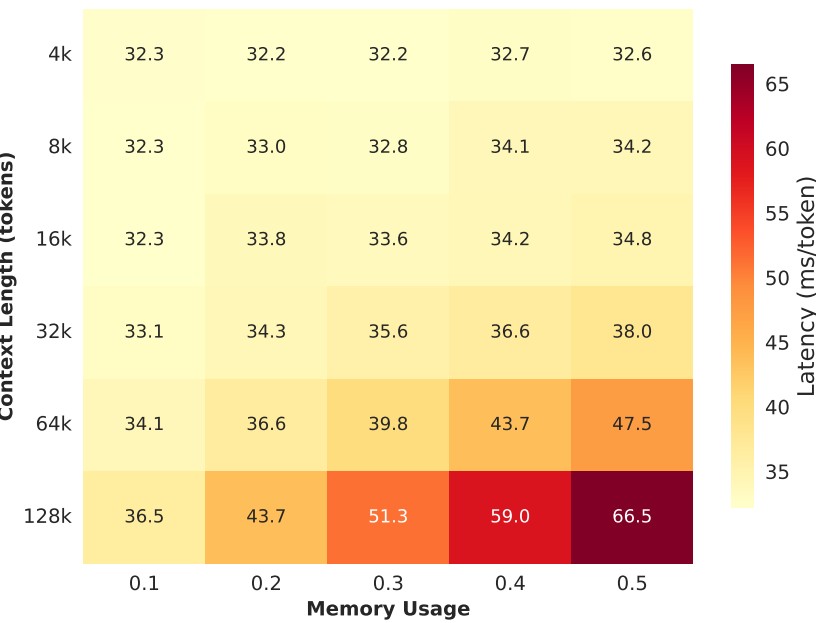

Figure 11: The generation latency of Llama-3.1-8B-Instruct on A6000.

