# OpenReview forum: "GVote: Adaptive Per-request KV-Cache Compression without Manually Setting Budget"
_ICLR.cc/2026/Conference — Submitted to ICLR 2026_

### Official Review · Reviewer_zbEY · 2025-10-16

**Soundness:** 2
**Presentation:** 2
**Contribution:** 3
**Rating:** 2
**Confidence:** 3

**Summary:**

This paper introduces GVote, a sampling-based method to determine the KV cache budget and the KV pairs to keep for efficient inference. With a couple minor hyperparameters, the authors show that GVote can automatically reduce memory and speed up inference on several tasks. While the algorithm seems sound, I mainly have some issues with the clarity and motivation.

**Strengths:**

1) Automatically determining the KV budget is an important problem, and this paper makes progress towards it.
2) Sizable memory reduction and better prefill latency scaling.

**Weaknesses:**

1) I think there is a lack of background/related work in the paper that makes the field seem too narrow. Given that the field of KV cache compression has matured over the past 2 years, there have been a lot more work than what the authors have included. For instance, many methods that use low-rank compression, sparse+low rank compression, or offloading KV caches to CPU are not touched upon, even though they all fall under the realm of KV caching algorithms. While the authors do not need to explicitly compare against these methods, I suggest a broader coverage of the literature.

2) The claim in Figure 2 that the curves look Gaussian is not convincing visually, since they mostly seem like simple unimodal distributions. A quantitative metric would make this claim seem less suspect.

3) Many (if not all) of the tasks are short generation tasks. To demonstrate the predictive power of GVote, it would better to show results on long generation tasks.

4) Various typos:
    - Line 129: backwards quotations
    - Line 351: KVCache should be two words
    - Line 365: We extensively tests -> We extensively test
    - Equation 1: Should the index for the keys and values go from 1 to t? KV pairs from current and previous tokens are used, not just previous tokens.

**Questions:**

1) Are models using chain of thought for GSM8K?

2) Is there a way to tune the dial to prioritize accuracy vs. efficiency?

3) Is there a way to constrain the efficiency? For example, if I want at least a 80% reduction in memory, is there a way to enforce this?

4) Do you have throughput metrics? Or does this optimize only for latency?

---

> ### Author Response · Authors · 2025-11-17
>
> Dear Reviewer zbEY,
>
> Thank you for your review. Your feedback highlights several important areas where we can significantly improve the paper's completeness and clarity. We'd like to discuss our revision plan with you to ensure it aligns with your expectations.
>
>
> Weakness
>
> 1. This is a very important point, and we agree that our literature review was too narrow. You are right that the field has matured significantly, and we should have covered the broader landscape of KV cache optimization.
> You mentioned various key directions, such as low-rank compression, sparse+low-rank methods, and offloading (and there are more directions). To ensure our revised section is truly comprehensive and gives proper credit to the field, **are there any key papers or specific works in these areas that you believe are essential for us to include and discuss?** Your expertise here would be a great guide for us.
> 2. Theoretically, due to the Law of Large Numbers, it must hold for very long sequences. May we ask for the specific criterion to verify "this is a Gaussian distribution" that you want to see?
> 3. This is a valid point about the scope of our experiments. Our method's primary strength is in compressing long prompts, which benefits each decoding step. For tasks where the generated text is extremely long, the task itself generally provides a short prompt(like reasoning). We will add a discussion on this to clarify our method's ideal use cases and limitations. Does this seem like a fair way to address this point?
> 4. Thank you for catching these specific typos and the error in Equation 1. We sincerely apologize for these mistakes and will correct all of them in the revision.
>
> Questions
> 1. We do not explicitly set it, but in logs we found the model automatically "think step by step" sometimes.
> 2. Increasing $p_{nuc}$ or $S$ would prioritize accuracy, while decreasing them favors efficiency.
> 3. As for the algorithm itself, no. But we could employ a simple engineering trick: re-allocate the cache with AdaKV using the minimal KV cache size.
> 4. Similar to the Point 5 reply to Reviewer jWPe, GVote is not currently optimized for throughput. As it uses a non-uniform KV cache layout, it's difficult to integrate into existing serving frameworks. This requires 1. writing a head-wise sparse attention kernel and 2. re-design the entire scheduling system because the kv cache layout is more granular. Meanwhile, we need to carefully tune the code to avoid additional overhead. For example, let's consider a trivial implementation of the scheduler, say if we schedule each "head" as a sequence.  This will lead to an explosion of the number of sequences (N -> L\*H\*N), which will make the scheduler *extremely* slow. It's a big challenge that beyond our capabilities.

---

> > ### Comment · Reviewer_zbEY · 2025-11-24
> >
> > Thanks for the clarifications. I have follow-up comments:
> >
> > 1. While it may not be necessary to compare to them, I recommend acknowledging the existence of compression methods that use low-rankness [1,2,3] or quantization [4, 5]. This is not a complete list.
> >
> > 2. Law of Large Numbers implies convergence to the mean, not convergence in distribution, so I'm not sure this is a valid claim. Something like showing Q-Q plots or fitting Gaussians to these curves would be more convincing.
> >
> > 3. Is there anything that functionally prevents evaluation on long generation tasks? Even if your method is optimized for long prompt and short generation settings, it would be nice to see how it performs on long generation even for 1 model/task.
> >
> > 4. It would be good to include your discussion on challenges with maximizing throughput in your paper.
> >
> > I will consider changing my score after I see a revised paper version that addresses my concerns here and in my previous comment.
> >
> > [1] LESS: https://arxiv.org/pdf/2402.09398
> >
> > [2] Loki: https://proceedings.neurips.cc/paper_files/paper/2024/file/1e027da6bec9ceb2ec37951ceeccae93-Paper-Conference.pdf
> >
> > [3] ShadowKV: https://arxiv.org/pdf/2410.21465
> >
> > [4] KIVI: https://arxiv.org/pdf/2402.02750
> >
> > [5] Palu: https://arxiv.org/pdf/2407.21118?

---

### Official Review · Reviewer_A2eh · 2025-10-26

**Soundness:** 3
**Presentation:** 2
**Contribution:** 2
**Rating:** 6
**Confidence:** 4

**Summary:**

The paper introduces an adaptive KV-cache compression scheme that operates per request. By moving beyond fixed-budget compression and choosing the compression ratio dynamically for each input, the approach targets lower memory and more efficient deployment without sacrificing accuracy.

**Strengths:**

* The motivation is clear and well argued, and the experiments are thorough and well documented.
* The distribution-aware synthetic-query mechanism is sound and convincing.

**Weaknesses:**

* The reliance on K from the last ground-truth query to stabilize selection is intuitive but under-justified. Please add theory/ablations showing when this helps vs. hurts.
* Figure 6 needs methodological clarity. If results are matched by accuracy, each method’s retained KV size / compression ratio and the corresponding latency/memory should be reported to ensure fair comparability.
* Minor: duplicated sentence at L256–L257.

**Questions:**

* Do you have empirical or theoretical evidence to justify using the K from the last ground-truth query?
* In Fig. 6, at the matched-accuracy operating point, what are each method’s KV-cache size (or compression ratio) and latency?

---

> ### Author Response · Authors · 2025-11-17
>
> Dear Reviewer A2eh,
>
> Thank you for your positive feedback and for the clear, actionable suggestions. It's very helpful to get such specific guidance. We'd like to confirm our plan for addressing your points.
>
> 1. To address your concern directly, we will conduct an ablation study. This study will compare our full GVote method against a variant that only uses the top-p selection from synthetic queries for voting, without the stabilization from the last real query to demonstrate its effectiveness.
> 2. You're right, Figure 6 needs more detail to ensure a fair comparison. We will report the detailed cache ratios in the following revision.  This is unrelated to the prefilling overhead analysis, only the generation overhead analysis.
> 3. Thank you for catching that! We will remove the duplicated sentence in our revision.
>
> Your suggestions are straightforward and will clearly improve the paper. Thank you again for your help.

---

### Official Review · Reviewer_jWPe · 2025-10-31

**Soundness:** 3
**Presentation:** 3
**Contribution:** 3
**Rating:** 4
**Confidence:** 4

**Summary:**

This paper addresses the limitations of fixed-budget KV-cache compression, where a single preset budget is applied to all requests regardless of their contextual complexity. This uniform setting can either hurt accuracy on difficult reasoning tasks or waste memory on simpler ones.

To resolve this, the authors propose GVote, an adaptive per-request KV-cache compression scheme that automatically determines the necessary cache budget with low overhead.
GVote is built on the observation that hidden states across the sequence approximately follow a Gaussian distribution. Leveraging this, the method samples synthetic future queries via Gaussian perturbation, aggregates their attention over existing keys through a Monte-Carlo voting procedure, and retains the union of key positions deemed important.

As a result, complex workloads (e.g., mathematical reasoning) naturally receive larger cache budgets, whereas simpler tasks (e.g., QA) are assigned smaller ones, enabling better accuracy–efficiency trade-offs. Experiments on Qwen2.5-7B-Instruct as well as Llama-based models across GSM8K, RULER, and LongBench demonstrate that GVote achieves higher or comparable accuracy at similar or lower memory usage compared to static-budget baselines.

**Strengths:**

* The paper clearly identifies the inefficiency of fixed-budget KV-cache compression under heterogeneous workloads and proposes a practical per-request budgeting mechanism.
* By combining Gaussian-based synthetic query generation with Monte Carlo voting to approximate future queries, the method offers a technically interesting and novel bottom-up perspective.
* The method demonstrates consistent accuracy–memory improvements across diverse benchmarks (e.g., GSM8K, RULER, LongBench) and model families (e.g., Qwen, Llama), supporting its generality.
* By dynamically estimating per-request budgets, the approach achieves higher accuracy at comparable memory usage, or similar accuracy with lower memory, compared to fixed-budget baselines.
* Sensitivity analyses on hyperparameters (e.g., p_nuc , S) provide useful guidance for applying the method in practice.

**Weaknesses:**

* The assumption that hidden states follow a Gaussian distribution is only weakly supported; stronger theoretical justification or broader empirical evidence is needed to establish generality.
* The paper provides limited analysis of failure cases in which synthetic queries fail to approximate future queries, leaving the robustness of the proxy insufficiently explored.
* Although the method claims to eliminate manual budget specification, several key hyperparameters still require tuning.
* Comparisons against recent adaptive KV-cache compression approaches are insufficient, making it difficult to assess the precise novelty of the contribution.
* The paper does not adequately discuss how the proposed non-uniform KV layout would integrate with real-world inference systems (e.g., paged attention in vLLM), where memory layout constraints and paging mechanisms may complicate deployment.
* The method focuses on single-request contexts and does not address how KV-cache should be maintained, shared, or incrementally updated across multi-turn dialogue scenarios, which are central in practical LLM applications.

**Questions:**

* Could you provide broader empirical evidence showing that the Gaussian assumption for hidden states holds consistently across different models, layers, and task types?
* In cases where synthetic queries fail to approximate future attention well, what conditions cause such failures, and how do they impact model performance?
* Could you provide a quantitative evaluation of run-to-run variance to assess the stability of the Monte-Carlo sampling process?
* How would the proposed non-uniform KV layout interact with paged-attention–based systems such as vLLM?
* How should the method maintain and update KV-cache across multi-turn dialogue scenarios, which are common in real LLM applications?
* Since the proposed method leverages hidden-state statistics rather than actual future tokens, would it be feasible to pre-compute a reusable pool of synthetic queries instead of re-sampling for every request?

---

> ### Author Response · Authors · 2025-11-17
>
> Dear Reviewer jWPe,
>
> Thank you for your constructive and detailed review. Your questions are helping us to think more deeply about our work. To ensure our revisions are as effective as possible, we would like to discuss our plan for addressing your points and ask for your input.
>
> 1. On the Gaussian Assumption (W1/Q1):
> You pointed out that the evidence for our Gaussian assumption needs to be stronger. You didn't state what specific evidence is desired. May we ask for the specific criterion to verify "this is a Gaussian distribution" that you want to see? We would provide a broader visualization indeed, since this is much specific.
>
> 2. On Failure Cases (W2/Q2):
> You asked for an analysis of cases where our synthetic queries fail. This is a great point. The term "failure" can be broad (low attention overlap? or low e2e scores? or else?), so we'd like to propose a concrete analysis: we can investigate the **attention heads where the attention overlap between our proxy and the ground truth is low, and analyze the properties of the corresponding context.** Does this specific, targeted analysis seem like a useful way to explore the "failure cases" you had in mind?
>
> 3. On Hyperparameters (W3):
> You are right that our method still has hyperparameters, and our claim to "eliminate manual specification" was imprecise. We will correct this. Our key point, which we will clarify, is that our hyperparameters (like p_nuc, S) are more about the process of finding the budget and are highly robust—**we used a single set of parameters across all varied benchmarks**. This is fundamentally different from a fixed budget, which needs to be re-tuned for each specific task or resource constraint. Does this clarification on the nature and robustness of our parameters address your concern?
>
> 4. On Comparison to Other Adaptive Methods (W4):
> You noted that our comparison to recent adaptive methods is insufficient. We agree, as the field is moving very (somewhat incredibly) quickly. To make our related work section more targeted, **are there any specific recent adaptive methods you believe are essential for us to discuss or compare against?**
>
> 5. On vLLM Integration and Multi-turn Dialogue (W5/Q4, W6/Q5):
> You've raised two crucial points about real-world applicability. We agree that systems integration is a major engineering challenge beyond our current scope, as is the extension to multi-turn dialogue. We will add a detailed discussion on these limitations. This leads to a question for us: from your perspective, setting aside these significant systems and application challenges, **how do you view the standalone novelty of the GVote algorithm itself? Or say, the concept of "adaptive budget"**? To us, it could gain a better trade-off between accuracy and cache size, yet the translation of this again into overall throughput is difficult, because of the potential overhead of a non-uniform attention kernel, and the total re-design of the scheduler and cache system.
>
> 6. On MC Stability (Q3):
>
> That's really a good suggestion. Although we haven't conducted formal experiments yet, a larger rollout size inherently reduces variance. We will report the variance analysis for a rollout size 8, and other rollout sizes (if we have enough spare capacity to complete all the promised experiments).
>
> 7. On pre-computation(Q6)
>
> Before we respond further, we'd like to ask what you mean by "instead of re-sampling for every request?"? To keep our (we and you) thinking within the same abstraction, let's briefly describe our understanding of the decoding process. For each request, it undergoes LLM computation, generating logits, sampling to generate the next token, and so on. Where is the "re"-sampling? And what do you want to suggest by "every request"? If you want to mean "sharing the statistics between different requests", the answer is potentially, and unfortunately, no.

---

### Official Review · Reviewer_RD3X · 2025-11-01

**Soundness:** 2
**Presentation:** 1
**Contribution:** 2
**Rating:** 2
**Confidence:** 4

**Summary:**

This paper proposes GVote, an adaptive KV Cache compression scheme that avoids manual budget setting.
GVote relies on LLM hidden states’ Gaussian distribution: it uses Monte-Carlo sampling to generate synthetic future queries, aggregates their needed keys to dynamically determine optimal cache budgets, and forms a keep-set via voting.
Experiments  show GVote reduces VRAM while keeping accuracy high or comparable to baselines (StreamLLM, SnapKV, AdaKV). It also has low generation latency and compatibility with modern kernels like FlashAttention.

**Strengths:**

+ The proposed method maintains its advantage across different architectures (LLaMA, Qwen) and various model sizes (3B-14B), demonstrating that it is not dependent on a specific model structure and has a broad range of applicability.

**Weaknesses:**

+ In the discussion of "ARE SYNTHESIZED QUERIES GOOD PROXIES?", there is no baseline for comparison. For instance, if tokens from the Prefix and Observation window, as used in SnapKV, are employed as proxies, what will the attention overlap be? This comparison is necessary to evaluate the effectiveness of synthesized queries as proxies.

+ The experimental results in the paper are questionable. According to the reported results, StreamLLM's performance is only slightly worse than SnapKV and AdaKV. However, based on the results in the AdaKV paper, StreamLLM’s performance is significantly worse than the other two methods. This discrepancy is reasonable, as StreamLLM only retains attention sinks and short-term information, which likely limits its overall performance compared to the other approaches.


+ The evaluation datasets used in the paper are not clearly specified. For example, the paper mentions evaluation on Longbench, but Longbench includes 21 different datasets, each with different metrics (such as Acc, F1, Rouge-L, Edit-Sim, etc.). The paper does not explain how the reported "accuracy" was derived.

+ Adaptive adjustment of the budget is not necessarily an advantage, as it also means the budget becomes uncontrollable. In practical scenarios, this could lead to unexpected OOM issues.

+ The paper has not been proofread, and there are numerous issues with the citation formatting.

**Questions:**

See above

---

> ### Author Response · Authors · 2025-11-17
>
> Dear Reviewer RD3X:
>
> Thank you for your review. We are genuinely grateful for such solid and insightful feedback. As we plan our revisions, we have a few points where your clarification would be incredibly helpful to ensure we're on the right track.
>
> 1. You suggested using SnapKV's prefix/obs window as a baseline. This was an important point for us, as it revealed that we failed to clearly explain our core technical challenge: our method aims to predict future queries, and due to causal attention, past tokens are not a direct proxy for this task.
>
> Your comment helped us refine our core claim. We will now state it more clearly as: "predicting the distribution of future tokens (in a limited range like 256, since average RoPE becomes too inaccurate for further positions) based on the distribution of past ones." To validate this reframed claim, we will add an ablation study focused on this predictive capability.  Does this reframing and corresponding validation seem like a more meaningful way to address your concern?
>
> 2. We have verified our results and observe that performance is highly sensitive to the benchmark configuration, a variance also seen in other reports (e.g., [kvpress infinitebench results](https://github.com/NVIDIA/kvpress/tree/main/evaluation/benchmarks#infinitebench)). **StreamLLM is indeed theoretically simpler, but not certainly practically worse**. Does this clarification help address the concern about the result's validity?
>
> 3. You're right, we lacked sufficient detail here. We will add a table to the appendix specifying the subsets used and our metric aggregation method. We strongly appreciate your advice on polishing this work!
>
> 4. This is a very insightful point about a core trade-off. Our approach prioritizes dynamic efficiency over the predictability of fixed budgets. We will revise the paper to explicitly discuss this "predictability vs. efficiency" design philosophy. Do you feel this discussion would adequately address this point?
>
> 5. Regarding the formatting, we used the standard ICLR template and are having trouble spotting the errors. To help us correct them, could you please provide one or two specific examples?
>
> Thank you again for this constructive dialogue. Your guidance is invaluable as we work to strengthen the paper!

---

### Meta-Review · Area_Chair_iEQL · 2025-12-30

**Summary:**

The paper proposes GVote, an adaptive KV-cache compression scheme. While the motivation is clear, reviewers raised critical concerns regarding missing baselines, the validity of the Gaussian assumption, and system integration. Crucially, the recommendation for rejection is driven by the authors' rebuttal strategy. Instead of providing concrete evidence or revised results, the authors predominantly responded with reverse questions and hypothetical plans. This approach failed to substantiate their claims or address the fundamental flaws identified during the review process.

**Reviewer Concerns:**

The rebuttal left major concerns outstanding as the authors sought negotiation rather than demonstration:

 - Validity of Baselines (Reviewer RD3X): Instead of providing the requested comparisons for synthesized queries, the authors merely asked if reframing their claim "would be meaningful," leaving the experimental validity unproven.
 - Gaussian Assumption (Reviewer jWPe, Reviewer zbEY): Both reviewers questioned the theoretical basis of the method. Rather than providing empirical validation (e.g., Q-Q plots), the authors asked reviewers to define the "specific criterion" needed for verification.
 - Related Work (Reviewer zbEY): The authors admitted missing literature but asked the reviewer to provide a list of essential papers rather than conducting the research themselves.

**Reviewer Scores:**

Since the authors offered questions instead of the required data, the reviewers' scores are unlikely to improve.
 - Reviewer RD3X (Score: 2): Remains Reject. The doubts regarding result validity and missing baselines were not resolved with actual data.
 - Reviewer jWPe (Score: 4): Likely remains Borderline Reject. The skepticism regarding the Gaussian assumption persists since no "broader empirical evidence" was presented.
 - Reviewer zbEY (Score: 2): Remains Reject. This reviewer explicitly conditioned a score change on seeing a "revised paper" with new data, which was not provided.
 - Reviewer A2eh (Score: 6): The score likely remains a 6 (Weak Accept).

---

### Decision · Program_Chairs · 2026-01-26

Reject